# Twist is the key to the gating of mechanosensitive ion channel NOMPC

**Jingze Duan[1], Chen Song[1,2]\***

[1]Center for Quantitative Biology, Academy for Advanced Interdisciplinary Studies, Peking University, Beijing, China; [2]Peking-Tsinghua Center for Life Sciences, Academy for Advanced Interdisciplinary Studies, Peking University, Beijing, China

## eLife Assessment

This study uses steered molecular dynamics simulations to interrogate force transmission in the mechanosensitive NOMPC channel, which plays roles including soft-touch perception, auditory function, and locomotion. The **valuable** finding that the ankyrin spring transmits force through torsional rather than compression forces may help understand the entire TRP channel family. The evidence is considered to be **solid**, although full opening of the channel is not seen, and it has been noted that experimental validation of reduced mechanosensitivity through mutagenesis of proposed ankyrin/TRP domain coupling interactions would help substantiate the findings.

**\*For correspondence:**
c.song@pku.edu.cn

**Competing interest:** The authors declare that no competing interests exist.

**Abstract** NOMPC, a tethered mechanosensitive ion channel belonging to the transient receptor potential (TRP) family, converts mechanical stimuli into ionic electric signals that excite neuronal cells (Yan et al., 2013). Previous investigations have demonstrated that a pushing force applied to the linker helix domain or the compression of NOMPC's ankyrin repeat (AR) domain can trigger channel opening (Wang et al., 2021). In this study, we explored the direct mechanical causes of NOMPC channel opening as well as the torsional properties of the AR domain, using all-atom molecular dynamics simulations. Our results indicate that a torque directed toward the extracellular side, exerted on the amphipathic TRP domain, is the primary factor driving channel opening. The coupling between compression and twisting of the AR domain ensures that both types of deformation can induce channel opening. Therefore, we propose a twist-to-open model, facilitated by the compression-twist coupling property of the AR domain, to provide further insight into the gating mechanism of the NOMPC channel.

## Introduction

Mechanosensitive (MS) ion channels are critical components of mechanosensory transduction, playing an essential role in the perception of sensory stimuli in living organisms. In both vertebrates and invertebrates, MS ion channels are widely expressed in peripheral sensory neurons located near or within the surface tissues responsible for detecting mechanical stimuli. Their primary function is to receive and transduce mechanical signals from the external environment, ultimately converting these signals into electrical impulses. This intricate process, known as mechano-electrical transduction (MET), induces changes in neuronal excitability, thereby facilitating the transmission of vital information to higher-order neural centers (*Hehlert et al., 2021*).

Among these, NOMPC, a transient receptor potential (TRP) superfamily MS channel identified in *Caenorhabditis elegans*, *Drosophila*, and zebrafish, plays a central role in various mechanosensing-related activities, including soft touch perception, auditory function, and locomotion (*Yan et al., 2013*; *Li et al., 2006*; *Effertz et al., 2011*; *Cheng et al., 2010*; *Sidi et al., 2003*). A salient characteristic

that distinguishes NOMPC from its counterparts is its unique intracellular domain, which comprises 29 ankyrin repeats (ARs) arranged into a distinctive spring-like conformation. This exceptional architecture, proposed to be associated with microtubules within cellular contexts, designates NOMPC as the only known standalone tether-gating MS channel (*Jin et al., 2017*). Consequently, in contrast to the extensively studied force-from-lipid MS ion channels (*Kefauver et al., 2020*; *Jin et al., 2020*), a detailed investigation into the specific molecular mechanism of NOMPC gating is poised to provide fundamental insights into the field of tethered MS ion channels.

In previous studies, it was proposed that compression of the intracellular AR domain, or a pushing force exerted on the AR domain from the intracellular side, could facilitate the opening of the channel (*Wang et al., 2021*; *Argudo et al., 2019*). Interestingly, the TRP domain, widely recognized as crucial in the gating process of TRP channels (*Liao et al., 2013*), was observed to tilt up when viewed along the membrane surface and rotate clockwise when viewed from the intracellular side (*Wang et al., 2021*). It appears that compression of the AR domain can generate both a membrane-normal pushing force and a membrane-parallel twisting force on the TRP domain (*Wang et al., 2021*; *Argudo et al., 2019*). However, it remains unclear whether the pushing force or the twisting torque acting on the TRP domain is the direct mechanical cause of channel gating. In this work, we determine the critical force component for the gating of NOMPC using all-atom molecular dynamics (MD) simulations. Our results suggest that the twisting force is the key factor in channel gating, while the compression-twist coupling of the AR domain ensures that the compression of the AR domain can also activate the channel.

## Results
### Membrane-parallel torsion force was observed on the TRP domain during the push-to-open gating process

For a comprehensive exploration of the atomistic gating details through MD simulations with an affordable computational cost, we employed a divide-and-conquer strategy as depicted in *Figure 1a*. Essentially, the membrane-protein simulation system was divided into two subsystems. System I was composed of the transmembrane (TM) region, intracellular linker helices (LH) domain, and AR 29. The TM region contains six TM helices (S1–S6) in each chain. Four S6 helices of each chain collectively constitute the central pore of NOMPC, and the TRP domain is sandwiched between the TM domain and the LH domain. System II included the LH domain and 29 AR units. The four spring-like AR chains of the tetrameric protein were organized into a bundled structure, which is believed to be a crucial element in mechanical perception (*Mosavi et al., 2004*; *Liang et al., 2013*; *Zhang et al., 2015*).

Building upon the previously proposed push-to-open model (*Wang et al., 2021*), we performed steered molecular dynamics (SMD) simulations for system I. In this context, a pushing/compressive force was applied to the AR29, pointing from the intracellular to the extracellular side (trajectories: CI-1/2). Consistent with our previous findings, the TM central pore of NOMPC exhibited a discernible dilation, accompanied by the upward tilt and clockwise rotation of the TRP domain (*Figure 1—figure supplement 1*). The upward tilt was probably caused by the membrane-normal component of the applied mechanical force, while the clockwise rotation was generated by the membrane-parallel component that led to a torque pointing to the extracellular side, raising the question of which of these two components was the direct driving force for the channel opening.

To address this pivotal question, we quantitatively analyzed the force distribution in the above simulations CI-1/2, so that we could decompose the force into the membrane-normal and membrane-parallel components to investigate which one could open the channel. In particular, we used the force distribution analysis (FDA) method (*Costescu and Gräter, 2013*) to quantify the interactions between pairwise residues along the force transmission pathway within system I (*Figure 1a*). According to our simulation condition and trajectory analyses, the force transmission initiated from the AR29 domain, traversed through the LH domain, and then proceeded to influence the geometrically adjacent yet non-covalently bonded TRP domain, ultimately guiding the S6 domain and instigating the pore opening. Our analysis showed that the strongest interactions between these domains were predominantly located between a few pairs of residues at the interfaces (*Figure 1—figure supplement 2*), including A1118-N1148, Y1109-D1142 on the AR29-LH interface, and K1244-E1571, D1236-R1581, L1256-S1577 on the LH-TRP interface. Their proximity in space made their Coulomb and

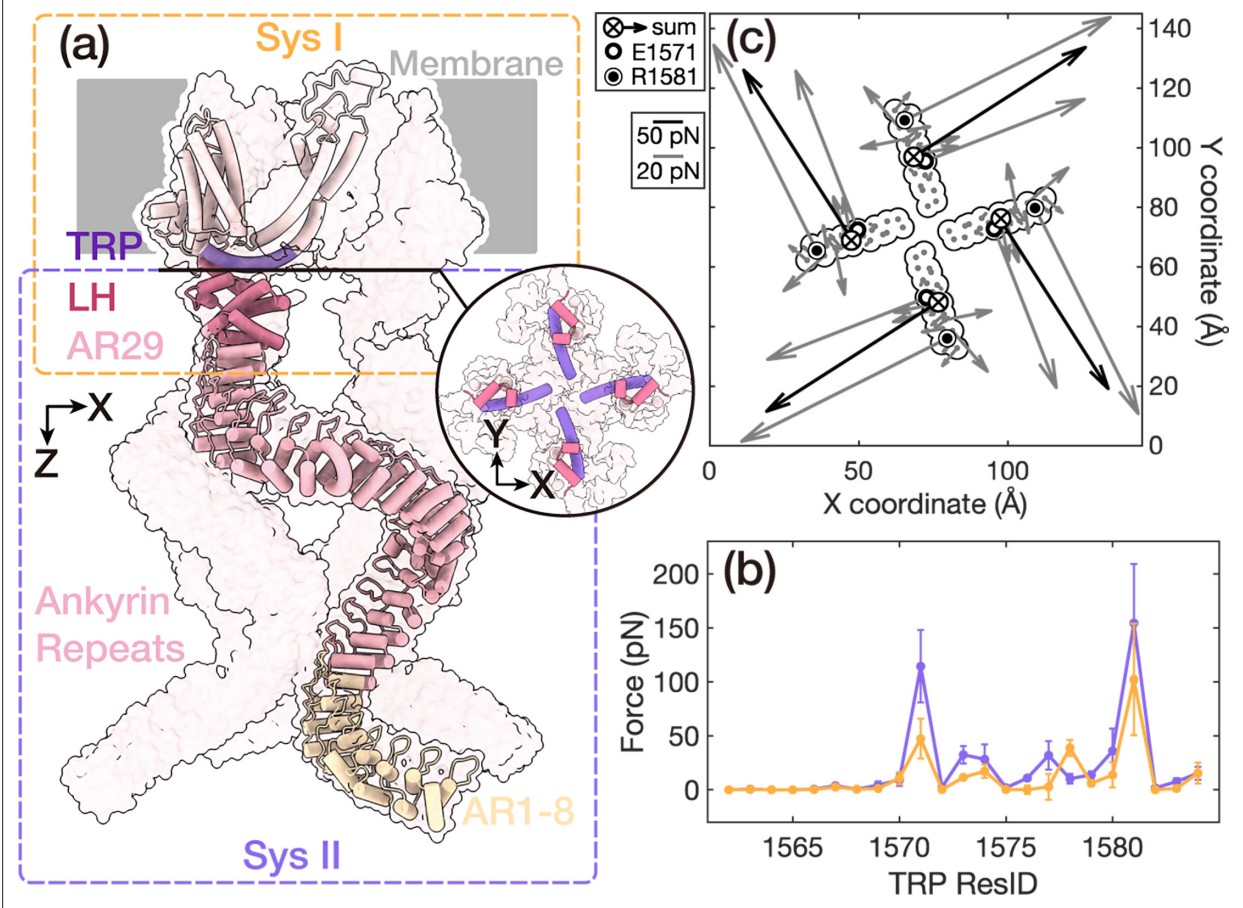

**Figure 1.** Force distribution during the push-to-open gating process of NOMPC. (**a**) The simulation systems of NOMPC. The NOMPC molecule was divided into two subsystems, denoted by the orange and purple rectangular boxes, for molecular dynamics (MD) simulations. The interface between the linker helices (LH) and the TRP domains is shown in a circular inset (bottom view). (**b**) The membrane-parallel (purple line) and membrane-normal (yellow line) components of the net forces on the TRP domain exerted by the LH domain. The error bars denote the standard deviations from two compressive trajectories (CI-1/2) and two free trajectories (FI-1/2). (**c**) The net forces on the TRP domain exerted by the LH domain. The gray quiver on each residue showed the net force exerted by the LH domain. The black shows the resultant force on the residues of the TRP domain. Two major points of force application, E1571 and R1581, are marked. ⊗ represents the inward force component. The data were averaged for the four subunits by rotationally symmetrized around the protein center.

The online version of this article includes the following figure supplement(s) for figure 1:

**Figure supplement 1.** Kinematic analysis of the push-to-open trajectory CI-1.

**Figure supplement 2.** The pairwise force distribution analysis (FDA) results of the push-to-open trajectory CI-1 between the AR29 domain, the LH domain, and the TRP domain.

**Figure supplement 3.** The force distribution analysis (FDA) results of the push-to-open trajectories between the LH domain and the TRP domain on the XY plane (TRP).

**Figure supplement 4.** The force distribution analysis (FDA) results of the push-to-open trajectory CI-1 between the TRP and transmembrane (TM) domain.

Lennard–Jones interactions stronger upon the conformational changes. This facilitated the stability of hydrogen bonds, particularly those between K1244-E1571, D1236-R1581, and L1256-S1577, with occupancies of 57%, 71%, and 24%, respectively. This finding was consistent with the hydrogen bond pairs identified in previous studies when a pushing force was applied (*Wang et al., 2021*).

To provide a clear depiction of the net mechanical force distribution resulting from the externally applied mechanical force, we subtracted the FDA results of the force-free simulation trajectories (trajectory: FI-1/2) from the simulation trajectories with pushing/compressive force (trajectory: CI-1/2). The residual force distribution was defined as the net-FDA results (*Figure 1—figure supplement 3*). By analyzing the net mechanical force distribution between the LH and TRP

domains using the net-FDA method, we observed that, in addition to the membrane-normal force component, the total net mechanical force transmitted from the LH domain to the TRP domain also exhibited a membrane-parallel torsion force component. This generated a torque directed toward the extracellular side, perpendicular to the membrane (*Figure 1b and c*). Moreover, the transmitted force was primarily concentrated on the E1571 and R1581 residues of the TRP domain (*Figure 1c*).

To determine the role of the TRP helix in gating, we analyzed the force transmission from the TRP domain to the TM region. The FDA results indicated that the TRP domain interacted with four TM segments, including portions of the S1 helix, the S2-S3 linker, the S4-S5 linker, and S6 (*Figure 1—figure supplement 4a, c, and d*). The strongest interaction occurred with a segment of the S6 domain that encompasses the constriction site of the TM pore (*Figure 1—figure supplement 4a*). Consequently, the TRP domain appears to exert an outward force on S6, potentially contributing to gating (*Figure 1—figure supplement 4b*). Furthermore, the TRP domain interacts strongly with D1360, located on the S2-S3 linker (*Figure 1—figure supplement 4a*), and with S1421 and F1422, located on the S4-S5 linker (*Figure 1—figure supplement 4a*). The presence of two pairs of hydrogen bonds between D1360 (S2-S3 linker) and K1578 (TRP), as well as S1421 (S4-S5 linker) and W1572 (TRP) was observed in our simulation and has also been reported in previous work (*Wang et al., 2021*; *Jin et al., 2017*).

The FDA results at the interface between the TM and intracellular domains elucidated a discernible pathway of force transfer, primarily through several residues located at the AR-LH-TRP interface. In the TRP domain, both membrane-normal and membrane-parallel force components were observed, providing a foundation for further investigation into which component serves as the primary factor in channel gating. As previous studies showed that a clockwise rotation of the TRP domain may be related to channel opening, it was of particular interest to study the effect of the torsion force on the channel structure.

## Membrane-parallel torsion force can open NOMPC, but not the membrane-normal pushing force on the TRP domain

Motivated by the above net-FDA results, we aimed to separate the contributions of the membrane-normal and membrane-parallel force components to the gating process and pinpoint whether one or both of these components played an indispensable role in gating. To achieve this, we needed to determine the point of application, the magnitude, and the direction of the exerted mechanical force. As the TRP domain is sandwiched between the AR and TM domains and its conformational change was proposed to be directly related to the channel gating, it is a natural choice to select the TRP domain to apply the decomposed forces in the MD simulations.

Notably, our FDA in the last section showed that the predominant forces propagated from the LH domain to the TRP domain were distributed on two key residues, E1571 and R1581 (*Figure 1b and c*). Therefore, we devised simulations to individually apply membrane-normal and membrane-parallel forces on these two residues of the TRP domain, and the magnitude and direction of the exerted forces were equivalent to those derived from the aforementioned net-FDA results (*Figures 1b and c and 2a and b*). As the force transmission occurred mainly via side-chain interactions along the interface between the LH and TRP domains, we selected the side chains of the E1571 and R1581 as the point of application of the exerted forces.

Our simulations revealed that the membrane-parallel torsion force alone was sufficient to open the gate in all three MD simulation replicates (trajectories: MPI-1/2/3, *Figure 2c*), whereas the membrane-normal pushing component did not trigger pore opening within the simulation time (trajectories: MNI-1/2/3, *Figure 2d*). According to the pore size evolution in the trajectories MPI-1/2/3 (*Figure 2c*), it became evident that the most constricted region along the pore, the gate situated around the hydrophobic I1554, exhibited an initial radius measuring less than 1.0 Å, as calculated by the HOLE program (*Smart et al., 1996*). This configuration represented a closed architecture. Under the influence of the membrane-parallel torsion force, the radius of the lower constriction was significantly dilated to above 2.0 Å after 600 ns. This dilation meant that water molecules could smoothly pass through the channel (*Figure 2—figure supplement 1*), a phenomenon often indicative of the first step of channel opening. This is highly similar to our previous observation when a pushing force was applied to the AR domain (*Wang et al., 2021*).

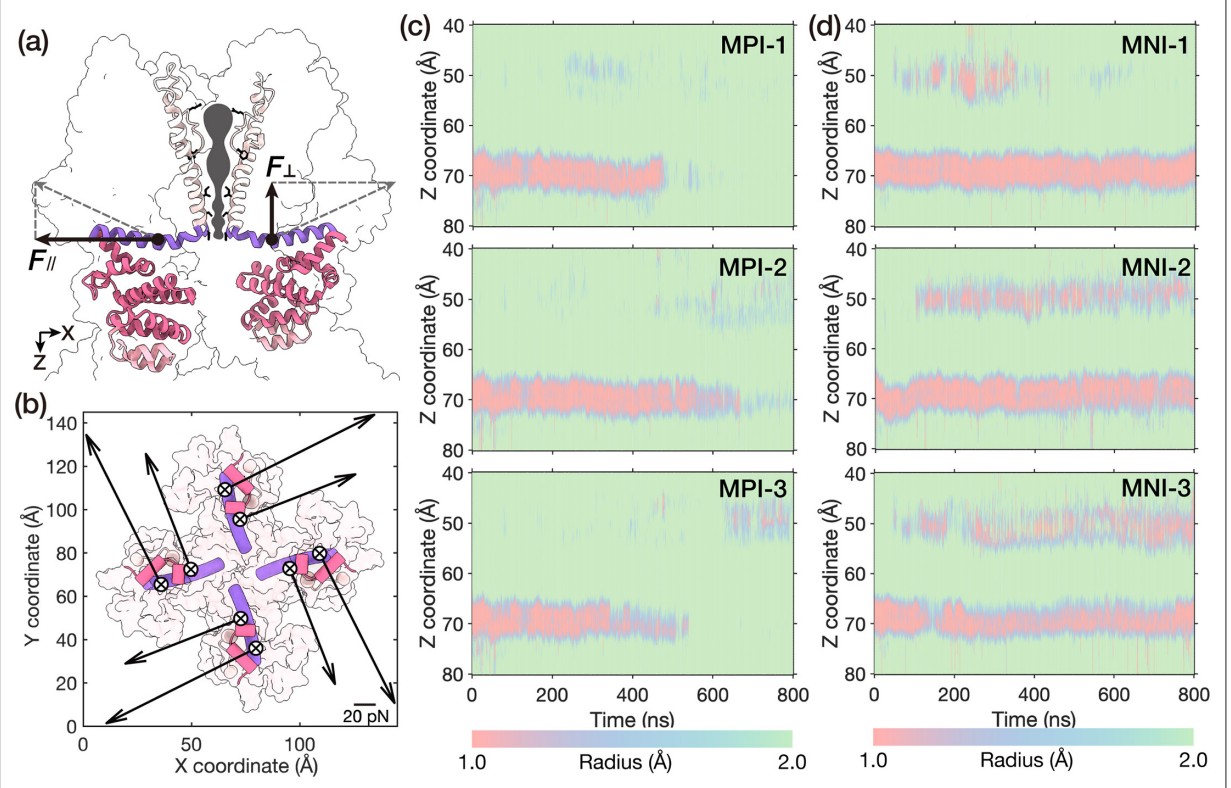

**Figure 2.** A membrane-parallel torsion force can open NOMPC. (**a, b**) The side view (**a**) and bottom view (**b**) of the net-FDA of the E1571 and R1581 on the TRP domain exerted from the LH domain, which was decomposed into the membrane-parallel and membrane-normal components. ⊗ represents membrane-normal forces pointing into the membrane. (**c, d**) The transmembrane pore size evolution in the simulations with membrane-parallel forces (**c**) or membrane-normal forces (**d**), three replicates for each condition (trajectories: MPI-1/2/3, MNI-1/2/3, see 'Methods').

The online version of this article includes the following figure supplement(s) for figure 2:

**Figure supplement 1.** Water molecules around the pore region.

**Figure supplement 2.** The transmembrane pore size evolution in the simulations with (**a**) half or (**b**) one-third magnitude of original membrane-parallel force, three replicates for each condition (trajectories: MPI-h1/2/3, MPI-t1/2/3).

**Figure supplement 3.** Secondary structure analysis of the TRP domain in the trajectory MPI-1.

In contrast, in the trajectories MN-1/2/3 with the membrane-normal force on the TRP domain, no pore dilation was observed (*Figure 2d*) within our simulation time. In fact, the channel appeared to become even more constricted at the upper constriction region (filter), as depicted in *Figure 2d*. This indicated that the membrane-normal force component on the TRP domain was probably not able to open the channel, at least no such sign was observed within our simulation time.

It should be noted that, according to our analysis, the membrane-normal pushing forces exerted on residues 1571 and 1581 are approximately one-third and two-thirds of the membrane-parallel twisting force, respectively. This observation indicates that the twisting component is inherently stronger than the pushing component when a compressive force is applied to the AR domain. To examine the impact of the two force components when they are of equal magnitudes, we conducted additional simulations (*Figure 2—figure supplement 2*, see 'Methods'). To ensure the stability of the protein and membrane structure, we opted to reduce the membrane-parallel force rather than increase the membrane-normal force. As illustrated in *Figure 2—figure supplement 2*, halving the membrane-parallel force resulted in channel opening in two out of three trajectories, while reducing it to one-third resulted in channel opening in one out of three trajectories. These results suggest that, when force components of equal magnitude are applied to the key residues on the TRP domain, the membrane-parallel (twisting) force is more effective in promoting channel opening than the membrane-normal (pushing) force.

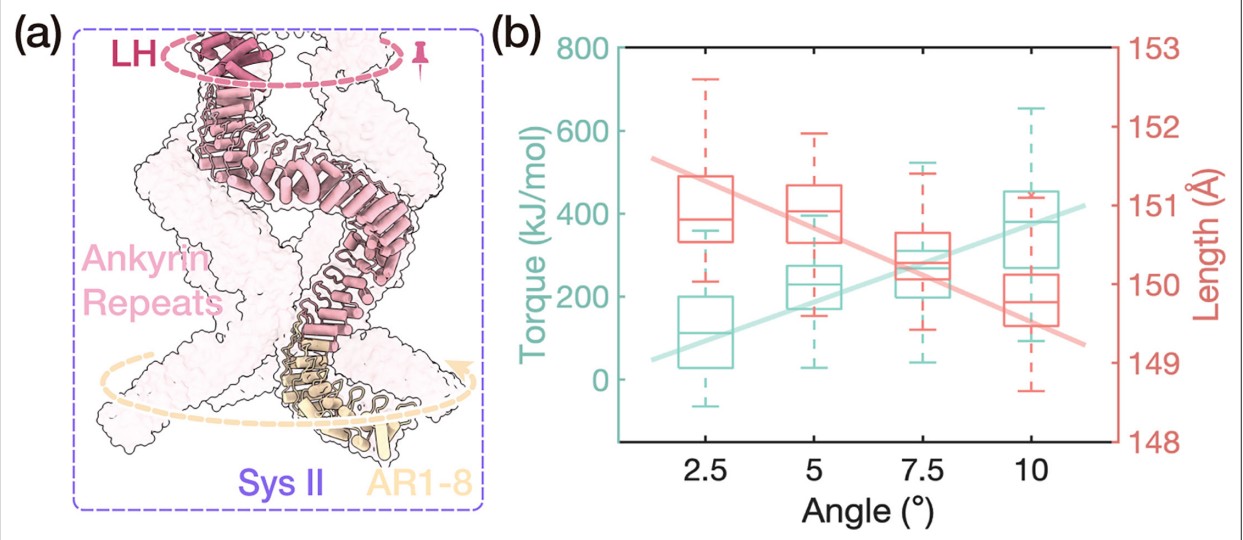

**Figure 3.** Torsional mechanical properties of the AR bundle. (**a**) The sketch of the steered molecular dynamics simulation. The dotted arrows reveal the direction of twisting. (**b**) The relationships between the torque on AR1-8 (left Y-axis), the length of the AR domain (right Y-axis), and the rotational angle of AR1-8. This was calculated from the rotating trajectories RII (0–4).

The online version of this article includes the following figure supplement(s) for figure 3:

**Figure supplement 1.** The torque (**a**) and rotational angle (**b**) versus time of trajectory RII, respectively.

**Figure supplement 2.** Torsional mechanical properties of the AR bundle.

Thus, our investigations substantiated that a torque on the TRP domain, pointing to the extra-cellular side, was the key to driving the channel open. On the basis of the original push-to-open or compress-to-open model, here we provide further details on the force analysis and channel opening, arguing that it is the membrane-parallel torsion force that acts as the direct mechanical cause to open the channel. It should be noted, though, this does not mean the push-to-open or compress-to-open model was wrong, as our simulations showed that a pushing force on the LH domain can generate a torsion force on the TRP domain (*Wang et al., 2021*), so does the compression of the AR domain (*Argudo et al., 2019*). Apparently, the unique structure of NOMPC ensures that a straightforward pushing force or compression of the AR spring can spontaneously generate a torsion force to open the channel. Therefore, the torsional mechanical properties of the AR domain became more intriguing.

### Torsional mechanical properties of the AR domain and force transmission pathway

To study the torsion property of the AR spring, we performed all-atom SMD simulations for system II, in which we enforced a gradual clockwise rotation to the lower end of the AR domain (AR1-AR8) at a rate of 0.05 °ns$^{-1}$ along the membrane-normal axis of NOMPC while keeping the upper end (LH domain) position-restrained (*Figure 3a*, trajectory: RII). From the trajectory, we extracted structural snapshots at an interval of 2.5 ° during the twisting process and relaxed these structures for 200 ns with both ends position-restrained. This protocol was employed to further relax the twisted AR spring to improve the accuracy of the calculated torsional mechanical parameters (*Figure 3—figure supplement 1*). Additionally, to ensure the convergence of the RII trajectory calculations, we simulated a set of trajectories with a faster application of the torsional force (trajectory: RII-fast). The results were highly consistent with those of RII (*Figure 3—figure supplement 2*). Please refer to the 'Methods section for more details.

With the above simulation data (trajectory: RII), we calculated the torsional coefficient of the AR bundle, using the equation:

$$M = c\varphi \tag{1}$$

where $M$ represents the torque applied to both ends, $c$ denotes the torsion coefficient of the AR domain bundle, and $\varphi$ is the average torsion angle of AR1-8 with respect to its initial position. As shown by the green plots in *Figure 3b*, the fitted torsion coefficient turned out to be $(2.30\pm0.31) \times 10^3$ kJ mol$^{-1}$ rad$^{-1}$. Such a torsion coefficient roughly corresponds to a similarly sized cylindrical volume of uniform polyethylene or rubber material (*Escudier and Atkins, 2019*).

Interestingly, we observed a gradual increase in the compressive deformation of the AR spring as the torsional angle increased in our simulations (*Figure 3b*). This implies a coupling between the twist and compression of the AR domain. To quantitatively characterize this coupling property, we defined the compression-twist coupling coefficient $k_{ct} = \frac{\Delta L}{\Delta \phi}$. This coefficient represents the compressive deformation induced by a unit torsional deformation. For the AR8-AR29 of NOMPC, the fitted value of the compression-twist coupling coefficient $k_{ct}$ was $(1.32\pm0.11)$ nm rad$^{-1}$ within a 10 ° torsion range. Indeed, it was also noticed that a torque could be generated during the application of compressive forces in previous investigations (*Wang et al., 2021*; *Argudo et al., 2019*). In light of these findings, we believe that the unique structure of the AR and LH domains enhances NOMPC's robustness in sensing external stimuli. Specifically, the compression-twist coupling property of the AR bundle

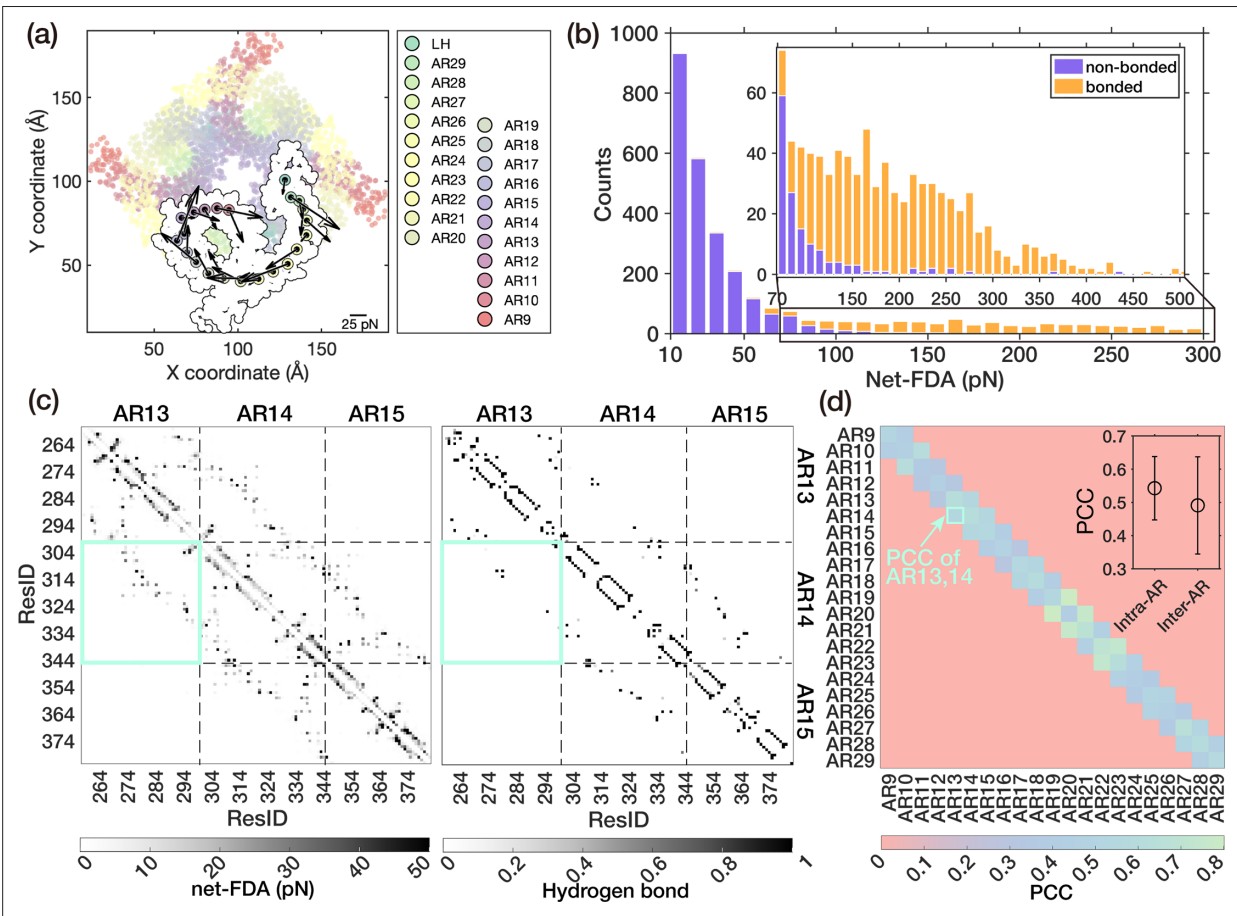

**Figure 4.** Force transmission along the AR domain. (**a**) The quiver plot of net-FDA (XY planar) of each $(i + 1)^{th}$ AR unit exerted from the $i$th AR unit shows the force transmission from the intracellular terminal to the transmembrane side of the AR domain viewing from AR1 toward AR29. $\odot$ represents the outward Z-component force. (**b**) The distribution of the pairwise residue-residue net-FDA within AR9 to AR29 (weak interaction under 5 pN neglected). The inset shows local information over 70 pN. (**c**) The representative interaction heatmaps (AR13-AR15) show the magnitude of non-bonded pairwise residue-residue net-FDA (left) and hydrogen bonds occupancy (right). (**d**) The Pearson correlation coefficient (PCC) between net-FDA and hbonds in AR units. The inset shows the mean value and standard deviation of Intra-AR (diagonal) and Inter-AR (sub-diagonal) PCCs.

The online version of this article includes the following figure supplement(s) for figure 4:

**Figure supplement 1.** Location of AR13-15 in the AR spring.

**Figure supplement 2.** Secondary structure analysis of the force-acting AR units in the trajectory RII-0.

ensures that both the twist and compression of the intracellular domain can generate torque to rotate the TRP domain, thereby facilitating the opening of the pore.

To pinpoint the force transmission pathway along the AR spring while it is twisted, we utilized the same net-FDA method as above to analyze the force distribution during the twisting process of the trajectory RII. The net-FDA results of each $(i + 1)^{th}$ AR unit exerted from the adjacent $i$th AR unit unveiled a spiral force transmission route, rooted in the helical structure inherent to the AR spring (*Figure 4a*). Strong pairwise interactions predominantly resided in bonded residue pairs, while only a small fraction was associated with non-bonded residue pairs (*Figure 4b*). Clearly, these bonded residue pairs, which are integral to the spiral backbone of the AR spring, were critical for transferring external forces. The non-bonded residue pairs were primarily situated at the adjacent turns within the AR α-helices and the interfaces between AR units. To clearly illustrate the details, we have magnified the local structure of AR13-15 (*Figure 4—figure supplement 1*). As depicted in *Figure 4c* -left, a representative non-bonded net-FDA of the local region of AR13-15 was shown. The data around the diagonal line represented the pairwise residues located at the adjacent turns between α-helices, while the wing-shaped data positioned further away from the diagonal indicated pairwise residues situated at the interfaces between AR units. Apparently, although the bonded residue pair interactions are predominant, the non-bonded interactions, both within the same AR unit and across different AR units, are important for the force transmission too.

Furthermore, we noticed that there is a correlation between the distribution of non-bonded net-FDA and the existence/strength of hydrogen bonds, as demonstrated by similar patterns in *Figure 4c*. For clarity, we divided these two 737 × 737 matrices (ResID of AR9-AR29: 400-1136) into 21 × 21 subgroups (grouped by AR unit). Then, we calculated the Pearson correlation coefficient (PCC) between the non-bonded net-FDA and the occupancy of hydrogen bonds of each subgroup, resulting in a 21 × 21 PCC matrix (*Figure 4d*). Relatively high PCCs in the sub-diagonal grids of *Figure 4d* (inter-AR), and even higher PCCs in the diagonal grids (intra-AR), are observed (inset of *Figure 4d*). This underscored the crucial role of the hydrogen bonding network in facilitating force transmission.

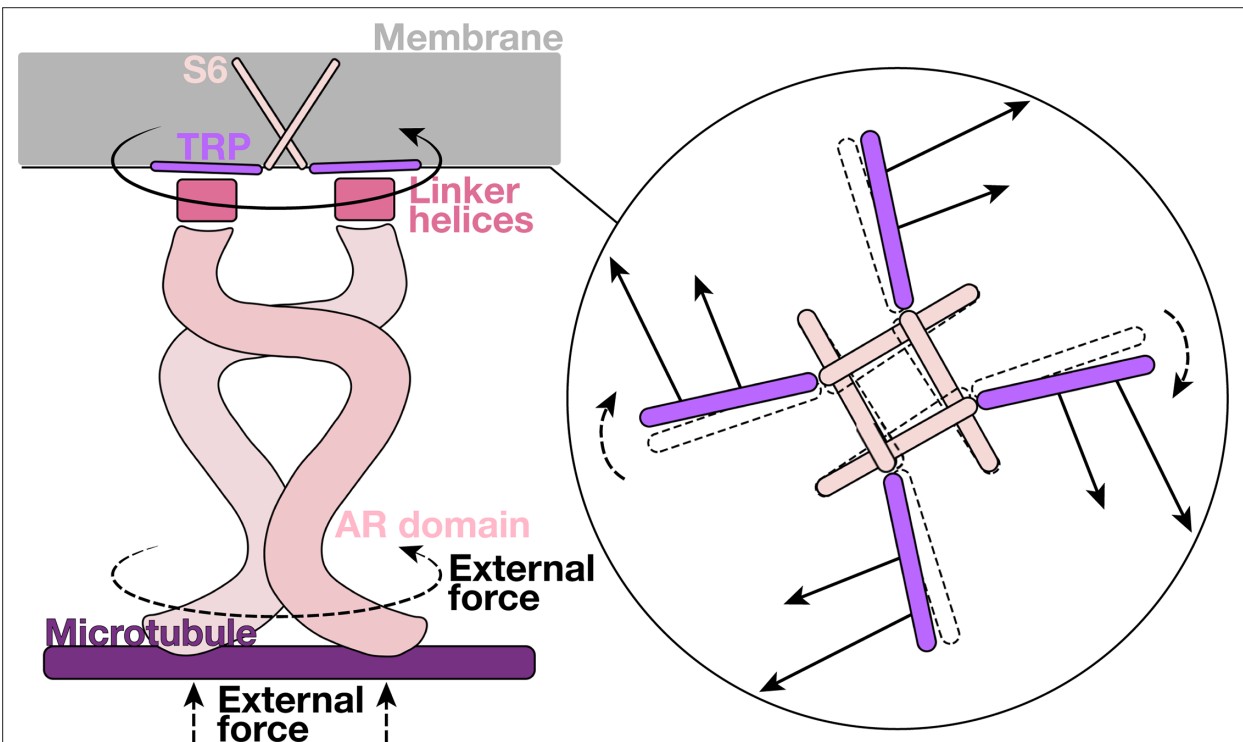

**Figure 5.** A schematic diagram of the twist-to-open model. The side view (left) and bottom view (right) of the twist-to-open model. Both the compression and clockwise twisting of the AR domain can generate a membrane-parallel twisting force on the TRP domain, which is the key component for gating.

Therefore, we conclude that the force transmission route primarily comprises bonded residues that constitute the spiral backbone of the AR chains, supplemented by the hydrogen bonding network formed by adjacent residues within the same or neighboring AR units. Although this may initially seem trivial, it offers valuable insights into the mechanism by which a twisting force can be formed and transmitted along the AR spring architecture. Furthermore, it underscores the stabilizing roles of both covalent and non-covalent interactions in preserving the stability of the force transmission pathway.

## Discussion

The unique structure of NOMPC and its fascinating AR spring have sparked great curiosity. Building upon the push-to-open and compress-to-open mechanism, here we propose a twist-to-open model, as illustrated in *Figure 5*. At the core of this model was the idea that a twisting force on the TRP domain is the key to the opening of the pore. This is not to say the compress-to-open or push-to-open models are wrong, but that a compression/pushing on the AR domain will lead to a twist force on the TRP domain, and this twist force is the direct cause of the pore opening. The spiral structure of each AR chain, as well as the hydrogen bonding network formed among spatially adjacent residues, enables the AR bundle to convert the compression deformation to a twist deformation, thus exerting a membrane-parallel torsional force on the LH domain. In addition, the structure around the LH domain can also convert a membrane-normal pushing force to a membrane-parallel torsional force, as shown in our previous study (*Wang et al., 2021*). Consequently, the compression of the AR domain, or a pushing force from the intracellular side, can be converted to a torque on the TRP domain, which points to the extracellular side. We believe that this torque is the direct mechanical cause that drives the TRP domain to rotate clockwise (looking from the intracellular side) to open the channel.

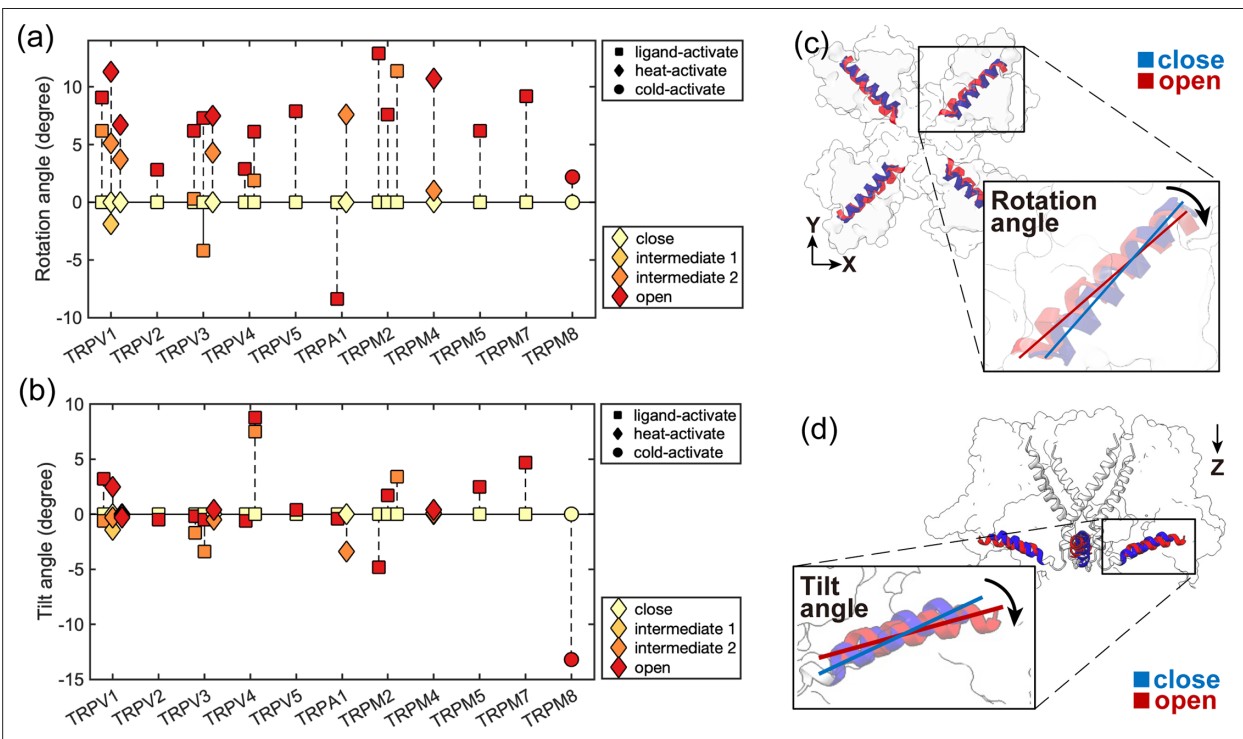

**Figure 6.** Conformational changes of the TRP domain in various TRP channels with resolved closed- and open-state cryo-EM structures. (**a**) The membrane-parallel rotational angle of the TRP helix (a positive value indicates clockwise rotation when viewed from the intracellular side). (**b**) The tilt angle of the TRP domain relative to the membrane surface (a positive value indicates tilt-up). S1-S4 helices were aligned for analysis. (**c**) and (**d**) represent the rotational and tilting conformational changes of the TRP domain upon gating, respectively. In the majority of the studied cases, the TRP domain rotates clockwise (**c**). The exception of counterclockwise rotation is observed for HsTRPA1. (Please refer to *Table 1* for the protein structures used in this analysis).

The online version of this article includes the following figure supplement(s) for figure 6:

**Figure supplement 1.** Rotation of the TRP domain of TRPV1 under torsional or pushing forces.

The highly conserved TRP domain serves as a structural characteristic of the TRP protein super-family. In our simulations of NOMPC, we observed a clockwise rotation of the TRP domain during the gating process of the proteins. Interestingly, such rotation was also observed in many other TRP channels. We investigated the likely conformational changes of multiple channels within the TRP family upon opening and found that the rotation of the TRP domain is a universal conformational change, whereas the tilt of the TRP domain is considerably less consistent (*Figure 6*). Based on these findings, we believe that the rotation of the TRP domain may be a common feature in the gating mechanism of these TRP channels, and the unique structure and compression-twist coupling property of the intracellular domain of NOMPC ensure its robustness in the perception of multiple forms of mechanical stimuli: not only twist, but also compression, can result in the clockwise rotation of the TRP domain, thereby facilitating channel opening.

Given the universal conformational change of the TRP domain and the preceding analysis of the interactions between the TRP helix and the TM region, it is pertinent to discuss the potential role of this amphipathic helix. As initially suggested in the MS channel MscL, researchers have identified that the amphipathic N-terminus of MscL provides a structural framework for bilayer-mediated gating of MS channels (*Bavi et al., 2016*). Subsequent studies have further demonstrated that many MS channels share a common structural feature: amphipathic helices that are directly or indirectly connected to pore-lining regions (*Kefauver et al., 2020*). These amphipathic helices interact with surrounding lipids and transmit membrane tension, which is consistent with the force-from-lipid gating mechanism. Although the present study primarily focuses on the force-from-tether model, it appears that the amphipathic TRP helix, which is connected to the pore-lining S6, is also crucial to the gating of NOMPC. The observed interaction between the TRP domain and the S4–S5 linker suggests that these two structural elements may function together as a composite amphipathic unit, potentially playing a significant role in perceiving and propagating gating signals. In addition, the TRP helix and the S4-S5 linker form the binding site for ligand-gated TRP channels, such as TRPV1. Therefore, we propose

**Table 1.** PDB ID of resolved cryo-EM structures used in *Figure 6*.

| Protein | State description | PDB ID |
|---|---|---|
| RnTRPV1 | apo→open (+RTX/DkTx) | 5IRZ, 7L2L, 5IRX; *Gao et al., 2016*; *Zhang et al., 2021* |
| RnTRPV1 | apo→open (+RTX,4°→25°C) | 5IRZ, 7RQU, 7RQX, 7RQZ; *Gao et al., 2016*; *Kwon et al., 2022* |
| RnTRPV1 | apo→open (+capsaisin,48°C) | 5IRZ, 7LPD, 7LPE; *Gao et al., 2016*; *Kwon et al., 2021* |
| RnTRPV2 | apo→open (+2APB,agonist) | 6U84, 7N0N; *Pumroy et al., 2019*; *Pumroy et al., 2022* |
| HsTRPV3 | apo→open (+THCV) | 8V6K, 8V6M, 8V6L; *Nadezhdin et al., 2024* |
| HsTRPV3 | apo→open (+2-APB) | 8V6K, 8V6O, 8V6N; *Nadezhdin et al., 2024* |
| MmTRPV3 | apo→open (42 °) | 7MIM, 7MIL, 7MIO; *Nadezhdin et al., 2021b* |
| HsTRPV4 | apo→open (+4a-PDD) | 8T1B, 8T1D; *Nadezhdin et al., 2023b* |
| HsTRPV4 | close→open (−antagonist, +agonist) | 8FC7, 8FC9, 8FCB; *Kwon et al., 2023* |
| OcTRPV5 | apo→open (+CoA) | 7T6J, 8FHI; *Fluck et al., 2022*; *Lee et al., 2023* |
| DmTRPA1 | close→sensitized (warm) | 7YKR, 7YKS; *Wang et al., 2023* |
| HsTRPA1 | close→open (inhibitor:A96; agonist:IA) | 6V9Y, 6V9X; *Zhao et al., 2020* |
| DrTRPM2 | apo→open (+ADPR, $Ca^{2+}$) | 6DRK,6DRJ; *Huang et al., 2018* |
| SrTRPM2 | apo→open (+ADPR, $Mg^{2+}$) | 8SRJ,8SRD; *Huang et al., 2024* |
| NvTRPM2 | close→pre-open (+ADPR, $Ca^{2+}$) | 6CO7,9JJE; *Zhang et al., 2018*; *Tóth et al., 2024* |
| HsTRPM4 | cold close→warm open (+DVT,$Ca^{2+}$) | 9B92, 9B8W, 9B8Y; *Hu et al., 2024* |
| DrTRPM5 | apo→open (+$Ca^{2+}$) | 7MBP, 7MBS; *Ruan et al., 2021* |
| MmTRPM7 | apo→open (+agonist) | 8SI2, 8SI5; *Nadezhdin et al., 2023a* |
| MmTRPM8 | apo→open (+cooling agonist) | 8E4P, 8E4L; *Yin et al., 2022* |

that the combined TRP-S4-S5 unit may constitute a key structural element responsible for multimodal gating mechanisms within the TRP channel family.

Mammalian TRPs possess shorter ARs than NOMPC, raising the question of whether they can also convey force in a manner similar to NOMPC. As an initial exploration, we selected the full-length squirrel TRPV1 (PDB: 7LQY) (*Nadezhdin et al., 2021a*) and applied pushing or torsional forces to the intracellular end of TRPV1, separately (*Figure 6—figure supplement 1*). Rotation of the TRP domain was observed under both types of mechanical stimulation (*Figure 6—figure supplement 1*). The torsional force-induced conformational change in the TRP domain resembles that observed in NOMPC. This suggests that a torsional force applied to the shorter ARs of mammalian TRPs may generate similar effects on channel gating. However, since these ARs are not tethered, the implications of these results in the context of biological functions remain unclear.

Although we calculated the detailed interactions between residue pairs of NOMPC, it should be noted that, due to the limitations of the FDA method, our force analysis lacked the entropic force, which may also play a significant role in the force transmission along the AR spring. Nonetheless, this won't affect our twist-to-open model, since the entropic effect was present in our MD simulations. Another limitation of the present study is that the simulation time is very short compared to the physiological time scale, so we only observed a partially open state of the structure in our simulations and cannot exclude the possibility that other factors, such as the membrane-normal force component, may also assist in the gating process. However, we found that the mechanical stimuli are transferred at a fast speed along AR springs in simulations, at around 1.8 nmps$^{-1}$ (*Wang et al., 2021*). Therefore, the mechanical stimuli were nearly instant for our simulation time, which allowed us to determine the trend or the direction of the conformational changes, even though the observation of the complete gating process is still challenging. In this context, we think that the dilation of the pore under a torsion force on the TRP domain is a piece of strong evidence that twist is the key to the channel opening.

## Methods
### All-atom simulation systems

To set up the MD simulations, we first prepared the simulation systems using the cryo-EM structure of NOMPC (PDB: 5VKQ) (*Jin et al., 2020*). CHARMM-GUI was used to generate the configuration and topology of the simulation systems (*Jo et al., 2009*). The parameter files were created for the CHARMM36m force field (*Lee et al., 2016*; *Huang et al., 2017*).

The system I included the TM domain, the LH domain, and the AR29 of NOMPC. The PPM server was used to reorient the NOMPC structure to ensure that the TM domain of NOMPC was well located in a lipid bilayer (*Huang et al., 2017*). The protein was embedded in a POPC bilayer. Then, we solvated the protein-membrane complex in a water box using the TIP3P water model. Counterions were added to neutralize the system, ensuring physiological ionic strength by adding Na$^+$ and Cl$^-$ ions to achieve a concentration of 0.15 M. The details of simulation system I are shown in *Table 2*.

The system II included the LH domain and the AR domain (AR1-AR29) of NOMPC. The protein structures were immersed in a rectangular box of TIP3 water molecules and 0.15 M NaCl. The details of simulation system II are shown in *Table 2*.

**Table 2.** Molecular dynamics simulation systems.

|  | System I | System II |
|---|---|---|
| Protein | TM, LH, AR29 | LH, AR1-AR29 |
| Box size | 145 × 145 × 145 Å$^3$ | 290 × 193 × 200 Å$^3$ |
| Atom number | 314,352 | 1,134,213 |
| Membrane | 492 POPC | / |
| Ions | 0.15 M NaCl | 0.15 M NaCl |
| Water number | 71,968 | 354,567 |

## All-atom MD simulations

All of the MD simulations were performed with GROMACS 5.1.2 (*Hess et al., 2008*). The REDUCE program in AMBER was used to add hydrogens to the original PDB files and determine the protonation state of the histidine residues (*Case et al., 2005*; *Word et al., 1999*).

For system I, we used the steepest descent algorithm to achieve energy minimization. Then, following up with a two-stage equilibration, a 0.4 ns NVT equilibration simulation with harmonic restraint was applied to the protein molecule (force constants of 4000 kJ mol$^{-1}$ nm$^{-2}$ on the backbone and 2000 kJ mol$^{-1}$ nm$^{-2}$ on the side chains), and a 20 ns NPT equilibration simulation with gradually decreased restraint (from 2000–100 kJ mol$^{-1}$ nm$^{-2}$ on the backbone and from 1000–50 kJ mol$^{-1}$ nm$^{-2}$ on the side chains). During the equilibration processes, planar restraints were used to keep the positions of lipid head groups along the membrane-normal direction. The simulation temperature of the system I was set to 300 K. The time step was 2 fs. The cubic periodic boundary condition was used during the simulations, and the van der Waals interaction was switched off from 10 to 12Å. The long-range electrostatic interactions were calculated using the particle mesh Ewald (PME) method (*Darden et al., 1993*). The Nosé–Hoover thermostat (*Nosé, 1984*) was applied separately to the protein, membrane, and solvent with a coupling constant of 1.0 ps and reference temperature of 300 K. Semiisotropic pressure coupling was applied using the Parrinello–Rahman algorithm (*Parrinello and Rahman, 1981*) with a reference pressure of 1 bar and a coupling constant of 5.0 ps. The compressibility was set to $4.5 \times 10^{-5}$ bar$^{-1}$. All bonds involving hydrogen were constrained using the LINCS algorithm (*Hess et al., 1997*).

For system II, we used the steepest descent algorithm to achieve energy minimization. Then, also followed by a two-stage equilibration, a 0.2 ns NVT equilibration simulation with harmonic restraint was applied to the protein molecule (force constants of 400 kJ mol$^{-1}$ nm$^{-2}$ on the backbone and 40 kJ mol$^{-1}$ nm$^{-2}$ on the side chains), and a 10 ns NPT equilibration simulation with harmonic restraint (force constants of 400 kJ mol$^{-1}$ nm$^{-2}$ on the backbone and 40 kJ mol$^{-1}$ nm$^{-2}$ on the side chains). The simulation temperature of the system II was set to 300 K. The time step was 2 fs. The cubic periodic boundary condition was used during the simulations, and the van der Waals interaction was switched off from

**Table 3.** Simulation trajectories of system I.

| Trajectory (label) | MPI-1/2/3 |
|---|---|
| Pulling group | Side chain of 1571 and 1581 |
| Force vector of each chain (kJ mol$^{-1}$ nm$^{-1}$) | On 1571: (3.4, 69.4, 0.0)<br>On 1581: (13.1, 91.5, 0.0) |
| Simulation time | 800 ns × 3 replicates |
| **Trajectory (label)** | **MPI-h1/2/3** |
| Pulling group | Side chain of 1571 and 1581 |
| Force vector of each chain (kJ mol$^{-1}$ nm$^{-1}$) | On 1571: (1.7, 34.7, 0.0) On 1581: (6.6, 45.8, 0.0) |
| Simulation time | 1000 ns × 3 replicates |
| **Trajectory (label)** | **MPI-t1/2/3** |
| Pulling group | Side chain of 1571 and 1581 |
| Force vector of each chain (kJ mol$^{-1}$ nm$^{-1}$) | On 1571: (1.1, 23.1, 0.0) On 1581: (4.4, 30.5, 0.0) |
| Simulation time | 1000 ns × 3 replicates |
| **Trajectory (label)** | **MNI-1/2/3** |
| Pulling group | Side chain of 1571 and 1581 |
| Force vector of each chain (kJ mol$^{-1}$ nm$^{-1}$) | On 1571: (0.0, 0.0, 28.3) On 1581: (0.0, 0.0, 61.2) |
| Simulation time | 800 ns × 3 replicates |

The force vectors are described in an X'Y'Z' coordinate system for each subunit. The positive direction of the X'-axis is oriented parallel to the TRP helix, pointing from the center of the pore towards the membrane. The positive direction of the Y'-axis is perpendicular to the TRP helix. The positive direction of the Z'-axis is perpendicular to the membrane, pointing towards the extracellular direction. The X'Y'Z' coordinate system follows the right-hand rule.

10 to 12 Å. The long-range electrostatic interactions were calculated using the PME method (*Darden et al., 1993*). The Nosé–Hoover thermostat (*Nosé, 1984*) was applied separately to the protein and solvent with a coupling constant of 1.0 ps and reference temperature of 300 K. Semiisotropic pressure coupling was applied using the Parrinello–Rahman algorithm (*Parrinello and Rahman, 1981*) with a reference pressure of 1 bar and a coupling constant of 5.0 ps. The compressibility was set to $4.5 \times 10^{-5}$ bar$^{-1}$. All bonds involving hydrogen were constrained using the LINCS algorithm (*Hess et al., 1997*).

### SMD simulations

SMD simulations were utilized to probe the mechanical gating mechanisms of the ion channels. SMD was performed by attaching a virtual spring to a specific group of atoms. To gradually apply a force to the designated group of atoms and prevent sudden conformational changes that could cause structural damage, a constant velocity was first applied to the spring to pull the molecule along a predefined reaction coordinate, followed by the application of a constant force. The force exerted on the molecule was recorded throughout the simulation. To ensure reproducibility, multiple independent SMD simulations were performed in this research.

The SMD simulation detail of the compressive trajectories of system I (CI-1/2) and free trajectories of system I (FI-1/2) can be found in previous work (*Wang et al., 2021*).

For trajectories MPI-1/2/3 (membrane-parallel force component on system I Replicate-1/2/3) and MNI-1/2/3 (membrane-normal force component on system I Replicate-1/2/3) of system I (*Figure 2c and d*), we separately exerted membrane-parallel (MPI-1/2/3) or membrane-normal (MNI-1/2/3) external force on the side chain of E1571 and R1581 (reference group: S1-S4), as shown in *Table 3*. At the first 200 ns, we applied an increasing force by using the constant-rate pulling method. The pulling was performed with a spring constant of 100 kJ mol$^{-1}$ nm$^{-2}$ and a pulling velocity of 0.005 nm s$^{-1}$. Then we applied the calculated constant force (*Table 3*) by using the constant-force pulling method in the last 600 ns. For additional simulations with smaller magnitude of membrane-parallel force component, we have trajectories MPI-h1/2/3 (half of membrane-parallel force component on system I Replicate-1/2/3) and trajectories MPI-t1/2/3 (one third of membrane-parallel force component on system I Replicate-1/2/3), as shown in *Table 3*.

Besides, to prevent the protein from rotation as a whole, we used the enforced rotation method (*Kutzner et al., 2011*), setting the rotation speed as zero for the S1-S4 helices in the TM region of the protein. The rotation potential was *rm2* with a force constant of 25 kJ mol$^{-1}$ nm$^{-2}$ and the pivot point for the potential was the center of the protein. The *norm* method was used to determine the angle of rotation group. Distance restraints were applied between the acceptors and donors on the side chains of residue pairs 1564–1568 and 1567–1571, to maintain the hydrogen bonds and prevent structural damage in the α-helix under the applied mechanical force.

The SMD simulation temperature of the system I was set to 300 K. The time step was 2 fs. The cubic periodic boundary condition was used during the simulations, and the van der Waals interaction was switched off from 10 to 12Å. The long-range electrostatic interactions were calculated using the PME method (*Darden et al., 1993*). The Nosé–Hoover thermostat (*Nosé, 1984*) was applied separately to the protein, membrane, and solvent with a coupling constant of 1.0 ps and reference temperature of 300 K. Semiisotropic pressure coupling was applied using the Parrinello–Rahman algorithm (*Parrinello and Rahman, 1981*) with a reference pressure of 1 bar and a coupling constant of 5.0 ps. The compressibility was set to $4.5 \times 10^{-5}$ bar$^{-1}$. All bonds involving hydrogen were constrained using the LINCS algorithm (*Hess et al., 1997*).

For trajectory RII (Rotation of system II), we performed a three-step simulation to better study the torsional mechanical properties of the AR domain, as shown in *Table 4*.

**Table 4.** Simulation trajectories of system II.

| Trajectory (label) | RII-0 | RII-1/2/3/4 | RII-fast-0 | RII-fast-1/2/3/4 |
|---|---|---|---|---|
| Rotation group 1 | LH | LH | LH | LH |
| Rotation group 2 | AR1-8 | AR1-8 | AR1-8 | AR1-8 |
| Rotational speed (°/ns) | 0.05 | 0 | 0.1 | 0 |
| Simulation time (ns) | 200 | 200 | 100 | 100 |

In the first step, we aimed to relax the structure by fixing the relative rotational angle between both ends along the rotational axis, which is the Z axis of system II. We choose to manipulate AR1-AR8 as a whole, considering the fact that the cryo-EM structure of AR1-AR7 was not resolved but modeled. Therefore, it was prudent to exert and transmit torque starting from the more reliably resolved structure, that is, from AR8 upward, while preserving the integrity of the entire protein structure. With the help of the enforced rotation function of GROMACS (*Kutzner et al., 2011*), we used the *pm* rotational potential on the bottom terminal of the AR domain (AR1-AR8) with a force constant of 100 kJ mol$^{-1}$ nm$^{-2}$ and a zero angular velocity for 20 ns, and the up terminal (LH domain) with a force constant of 50 kJ mol$^{-1}$ nm$^{-2}$ and a zero angular velocity for 20 ns. The *pm* rotational potential can constrain the group of atoms to rotate within the rotational plane as required, while allowing free movement in the direction perpendicular to the rotational plane. This means that the entire AR bundle will be subjected to torsion, while being able to stretch and compress freely. The pivot point for the potential was the center of the protein. The norm method was used to determine the angle of the rotation group.

In the second step, we clockwise rotated the AR1-8 along the Z axis at an angular velocity of 0.05° ns$^{-1}$ with a force constant of 100 kJ mol$^{-1}$ nm$^{-2}$ and, while keeping the LH domain still (*Figure 3a*) with *pm* rotational potential with a force constant of 50 kJ mol$^{-1}$ nm$^{-2}$. The difference in rotation angle between the bottom and up terminal gradually increased to 10° over 200 ns (*Figure 3—figure supplement 1*, trajectory: RII-0). The pivot point for the potential was the center of the protein. The norm method was used to determine the angle of the rotation group.

In the third step, when rotating at every 2.5° in trajectory RII-0, we extracted structural snapshots and held both ends of the AR domain for 200 ns to get a better equilibrated structure (*Figure 3—figure supplement 1*, trajectory: RII-1 for the 2.5°-rotational angle, RII-2 for 5°, RII-3 for 7.5°, and RII-4 for 10°), and the readouts of torques on both ends gradually increased with opposite directions. The other parameters for enforced rotation were kept consistent with those used in the previous two steps. Specifically, we used the trajectory RII-1/2/3/4 to calculate the torsional mechanical properties of the AR bundle and the trajectory RII-0 to study the force transmission pathway along the AR chain. When the torsional angles of the AR bundle are 2.5$i$° ($i$=1, 2, 3, 4), we analyze the 200 ns-long RII-$i$ trajectories. For the torques at both ends of the AR, we collect torque data on the LH domain every 2 ps. Under equilibrium conditions, the torques at the top and bottom of the AR bundle are equal in magnitude but opposite in direction. To assess the length change of the AR bundle, we measure the distance between the centroids of the LH domain and AR8 unit.

To further investigate the convergence of the torsional mechanical properties of the AR bundle and better compare the results with trajectory RII, we conducted additional simulations with a faster velocity of force loading to ensure robustness and consistency. We adopted the same first step as in trajectory RII. The rotational speed in the second step was doubled to 0.1° ns$^{-1}$, resulting in a total twist of 10° within 100 ns. The holding time in the third step was adjusted to 100 ns. This accelerated simulation trajectory is referred to as RII-fast (see *Table 4*). All other simulation parameters remained unchanged. The mechanical properties derived from the RII and RII-fast trajectories were found to be close (see SI), indicating that the results from the RII trajectory set are converged.

The SMD simulation temperature of the system II was set to 300 K. The time step was 2 fs. The cubic periodic boundary condition was used during the simulations, and the van der Waals interaction was switched off from 10 to 12 Å. The long-range electrostatic interactions were calculated using the PME method (*Darden et al., 1993*). The Nosé–Hoover thermostat (*Nosé, 1984*) was applied separately to the protein and solvent with a coupling constant of 1.0 ps and reference temperature of 300 K. Semiisotropic pressure coupling was applied using the Parrinello–Rahman algorithm (*Parrinello and Rahman, 1981*) with a reference pressure of 1 bar and a coupling constant of 5.0 ps. The compressibility was set to $4.5 \times 10^{-5}$ bar$^{-1}$. All bonds involving hydrogen were constrained using the LINCS algorithm (*Hess et al., 1997*).

We used Define Secondary Structure of Proteins (DSSP) analysis to ensure the stability of the secondary structure of NOMPC when applying SMD, for the TRP domain in system I and the AR units in system II (*Figure 2—figure supplement 3*, *Figure 4—figure supplement 2*).

## Force distribution analysis

FDA focuses on computational efficiency and low-memory usage and makes possible time series analysis of pairwise forces variation in long MD simulations and for large molecular systems (*Costescu*

*and Gräter, 2013*). It can compute atomic pairwise forces for all types of bonded interactions as well as Coulomb and Lennard–Jones potential by rerunning the trajectories. By summing up all the atomic pairwise forces within the two residues of interest, it can also calculate the pairwise force of the interaction between residues $r_i$ and $r_j$, which is labeled as $\vec{F}_{r_i,r_j}$. Similarly, if we focus on the interaction between two domains, we can sum up all the pairwise forces of the residues from these two domains.

In the calculation, we made an input file which contained all the parameters that FDA needed to run. We set `onepair = detailed` and `type = all` so that all pairwise forces of different types between the same pair of residues are stored separately. The option `residuebased` was set as `pairwise_forces_vector` so that the residue-based pairwise forces were written to output files as vector values. The forces for every frame were written by setting `time_averages_period = 0`.

Then, forces sampled by MD simulations under two conditions (with and without perturbation) were compared. We analyzed the interactions between residues in the trajectories with the external force (CI-1/2), labeled as $\vec{F}_{r_i,r_j,\text{CI-1}}$, $\vec{F}_{r_i,r_j,\text{CI-2}}$, and without the external force (FI-1/2), labeled as $\vec{F}_{r_i,r_j,\text{FI-1}}$, $\vec{F}_{r_i,r_j,\text{FI-2}}$, respectively. Then the average net force distribution induced by the compressive external force, which was named net-FDA, $\overline{\vec{F}_{r_i,r_j,\text{net}}}$, was obtained by subtracting the FDA result of trajectories without the external force from the FDA result of trajectories with the external force,

$$\overline{\vec{F}_{r_i,r_j,\text{net}}} = \frac{\vec{F}_{r_i,r_j,\text{CI-1}} + \vec{F}_{r_i,r_j,\text{CI-2}} - (\vec{F}_{r_i,r_j,\text{FI-1}} + \vec{F}_{r_i,r_j,\text{FI-2}})}{2}. \tag{2}$$

In the end, we averaged the vector net-FDA results of the four subunits by rotationally symmetrized around the protein center.

## Supplementary MD simulations of TRPV1

To conduct additional MD simulations of TRPV1, we prepared the simulation systems using the high-resolution cryo-EM structure of sqTRPV1 (PDB: 7LQY) (*Nadezhdin et al., 2021a*) which comprises six AR units (*Figure 6—figure supplement 1*). CHARMM-GUI was utilized to generate the configuration and topology of the simulation systems (*Jo et al., 2009*) and the CHARMM36m force field was used for the simulations (*Huang et al., 2017*).

The protein system consisted of the AR domain (AR1-AR6), the Linker domain, the Pre-S1 domain, the TM domain (TM domain, including the S1-S6 helix), the TRP domain, and the C-terminal of TRPV1. The PPM server was employed to reorient the TRPV1 structure, ensuring that the TM domain was properly inserted into a POPC lipid bilayer (*Lomize et al., 2012*). Then, we solvated the protein-membrane complex in a water box using the TIP3P water model. Counterions were added to neutralize the system and achieve physiological ionic strength by adding $Na^+$ and $Cl^-$ ions to reach a concentration of 0.15 M. The size of the box is 164 × 164 × 167 Å and the system contains 462,787 atoms.

All of the supplementary MD simulations were performed with GROMACS 2023.1 (*Hess et al., 2008*). The REDUCE program within AMBER was utilized to add hydrogen atoms to the original PDB files and to determine the protonation states of the histidine residues (*Case et al., 2005*; *Word et al., 1999*). We employed the steepest descent algorithm for energy minimization. Subsequently, a two-stage equilibration was conducted, which included a 0.4 ns NVT equilibration simulation with harmonic restraints applied to the protein molecule (with force constants of 4000 kJ mol$^{-1}$ nm$^{-2}$ for the backbone and 2000 kJ mol$^{-1}$ nm$^{-2}$ for the side chains), followed by a 20 ns NPT equilibration simulation with gradually decreasing restraints (ranging from 2000–100 kJ mol$^{-1}$ nm$^{-2}$ for the backbone and from 1000–50 kJ mol$^{-1}$ nm$^{-2}$ for the side chains). During the equilibration processes, planar restraints were implemented to maintain the positions of lipid head groups along the membrane-normal direction. The simulation temperature of the system I was set to 300 K. The time step was 2 fs. The cubic periodic boundary condition was used during the simulations, and the van der Waals interaction was switched off from 10 to 12 Å. The long-range electrostatic interactions were calculated using the PME method (*Darden et al., 1993*). The v-rescale thermostat was applied separately to the protein, membrane, and solvent with a coupling constant of 1.0 ps and reference temperature of 300 K (*Bussi et al., 2007*). Semiisotropic pressure coupling was applied using the c-rescale algorithm with a reference pressure of 1 bar and a coupling constant of 5.0 ps (*Bernetti and Bussi, 2020*). The compressibility was set to 4.5 × 10$^{-5}$ bar$^{-1}$. All bonds involving hydrogen were constrained using the LINCS algorithm.

For SMD simulations involving compressive force, a membrane-normal pushing force of 20 pN was applied to the AR1-2 units of each chain over a duration of 500 ns (*Figure 6—figure supplement 1*). In the case of SMD with rotational force, enforced rotation was implemented, whereby the AR1-2 units were rotated clockwise (when viewed from the intracellular side) around the center of TRPV1 along the Z-axis at an angular velocity of $0.0025°\ ns^{-1}$ with rotational potential pm at the force constant $25\ kJ\ mol^{-1}\ nm^{-2}$. During this process, the S1-S4 domain was held stationary through the application of the *pm* rotational potential at the force constant $25\ kJ\ mol^{-1}\ nm^{-2}$. This resulted in a rotation angle of 10° accumulated over 400 ns (*Figure 6—figure supplement 1d, e*). The SMD simulation temperature of the system was set to 300 K. The time step was 2 fs. The cubic periodic boundary condition was used during the simulations, and the van der Waals interaction was switched off from 10 to 12 Å. The long-range electrostatic interactions were calculated using the PME method (*Darden et al., 1993*). The v-rescale thermostat was applied separately to the protein, membrane, and solvent with a coupling constant of 1.0 ps and reference temperature of 300 K (*Bussi et al., 2007*). Semiisotropic pressure coupling was applied using the c-rescale algorithm with a reference pressure of 1 bar and a coupling constant of 5.0 ps (*Bernetti and Bussi, 2020*). The compressibility was set to $4.5 \times 10^{-5}\ bar^{-1}$. All bonds involving hydrogen were constrained using the LINCS algorithm.

## Acknowledgements

We would like to express our gratitude to Dr. Yang Wang at Wenzhou Institute, University of Chinese Academy of Sciences, for his assistance in building the initial simulation systems and for his insightful discussions. We also extend our appreciation to Dr. Jie Yan and his team at the National University of Singapore for their dedicated efforts in experimental exploration. This work was supported by the General Program of the National Natural Science Foundation of China (Grant No. 32071251), and the Science Fund for Innovative Research Groups of the National Natural Science Foundation of China (Grant No. T2321001).

## Additional information

### Funding

| Funder | Grant reference number | Author |
| --- | --- | --- |
| National Natural Science Foundation of China | 32071251 | Chen Song |
| National Natural Science Foundation of China | T2321001 | Chen Song |

The funders had no role in study design, data collection and interpretation, or the decision to submit the work for publication.

### Author contributions

Jingze Duan, Data curation, Formal analysis, Validation, Investigation, Visualization, Writing – original draft, Writing – review and editing; Chen Song, Conceptualization, Resources, Supervision, Funding acquisition, Validation, Investigation, Methodology, Writing – original draft, Project administration, Writing – review and editing

### Author ORCIDs

Jingze Duan ⓘ https://orcid.org/0009-0008-4677-7583
Chen Song ⓘ https://orcid.org/0000-0001-9730-3216

Reviewer #1 (Public review): https://doi.org/10.7554/eLife.102941.3.sa1
Reviewer #2 (Public review): https://doi.org/10.7554/eLife.102941.3.sa2
Reviewer #3 (Public review): https://doi.org/10.7554/eLife.102941.3.sa3
Author response https://doi.org/10.7554/eLife.102941.3.sa4

# Additional files

## Supplementary files
MDAR checklist

## Data availability
The current manuscript is a computational study. The main simulation setup and trajectory files have been uploaded to Zenodo (https://doi.org/10.5281/zenodo.17130035).

The following dataset was generated:

| Author(s) | Year | Dataset title | Dataset URL | Database and Identifier |
|---|---|---|---|---|
| Duan J | 2025 | MD simulation data for "Twist is the key to the gating of mechanosensitive ion channel NOMPC" | https://zenodo.org/records/17130035 | Zenodo, 10.5281/zenodo.17130035 |

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
