## [Editor Report · eLife Assessment]

This study uses steered molecular dynamics simulations to interrogate force transmission in the mechanosensitive NOMPC channel, which plays roles including soft-touch perception, auditory function, and locomotion. The **valuable** finding that the ankyrin spring transmits force through torsional rather than compression forces may help understand the entire TRP channel family. The evidence is considered to be **solid**, although full opening of the channel is not seen, and it has been noted that experimental validation of reduced mechanosensitivity through mutagenesis of proposed ankyrin/TRP domain coupling interactions would help substantiate the findings.

---

## [Referee Report · Reviewer #1 (Public review)]

Summary:

This manuscript uses molecular dynamics simulations to understand how forces felt by the intracellular domain are coupled to opening of the mechanosensitive ion channel NOMPC. The concept is interesting - as the only clearly defined example of an ion channel that opens due to forces on a tethered domain, the mechanism by which this occur are yet to be fully elucidated. The main finding is that twisting of the transmembrane portion of the protein - specifically via the TRP domain that is conserved within the broad family of channels- is required to open the pore. That this could be a common mechanism utilised by a wide range of channels in the family, not just mechanically gated ones, makes the result significant. It is intriguing to consider how different activating stimuli can produce a similar activating motion within this family. While the authors do not see full opening of the channel, only an initial dilation, this motion is consistent with partial opening of structurally characterized members of this family.

Strengths:

Demonstrating that rotation of the TRP domain is the essential requirement for channel opening would have significant implcaitions for other members of this channel family.

Weaknesses:

The manuscript centres around 3 main computational experiments. In the first, a compression force is applied on a truncated intracellular domain and it is shown that this creates both a membrane normal (compression) and membrane parallel (twisting) force on the TRP domain. This is a point that was demonstrated in the authors prior eLife paper - so the point here is to quantify these forces for the second experiment.

The second experiment is the most important in the manuscript. In this, forces are applied directly to two residues on the TRP domain with either a membrane normal (compression) or membrane parallel (twisting) direction, with the magnitude and directions chosen to match that found in the first experiment. Only the twisting force is seen to widen the pore in the triplicate simulations, suggesting that twisting, but not compression can open the pore. This result is intriguing and there appears to be a significant difference between the dilation of pore with the two force directions. When the forces are made of similar magnitude, twisting still has a larger effect than forces along the membrane normal.

The second important consideration is that the study never sees full pore opening, rather a widening that is less than that seen in open state structures of other TRP channels and insufficient for rapid ion currents. This is something the authors acknowledge in their prior manuscript Twist may be the key to get this dilation, but we don't know if it is the key to full pore opening. Structural comparison to open state TRP channels supports that this represents partial opening along the expected pathway of channel gating.

Experiment three considers the intracellular domain and determines the link between compression and twisting of the intracellular AR domain. In this case, the end of the domain is twisted and it is shown that the domain compresses, the converse to the similar study previously done by the authors in which compression of the domain was shown to generate torque.

---

## [Referee Report · Reviewer #2 (Public review)]

This study uses all atom MD simulation to explore the mechanics of channel opening for the NOMPC mechanosensitive channel. Previously the authors used MD to show that external forces directed along the long-axis of the protein (normal to the membrane) results in AR domain compression and channel opening. This force causes two changes to the key TRP domains adjacent to the channel gate: (1) a compressive force pushes the TRP domain along the membrane normal, while (2) a twisting torque induces a clock-wise rotation on the TRP domain helix when viewing the bottom of the channel from the cytoplasm. Here, the authors wanted to understand which of those two changes are responsible for increasing the inner pore radius, and they show that it is the torque. The simulations in Figure 2 probe this question with different forces, and we can see the pore open with parallel forces in the membrane, but not with the membrane-normal forces. I believe this result as it is reproducible, the timescales are reaching 1 microsecond, and the gate is clearly increasing diameter to about 4 Å. This seems to be the most important finding in the paper, but the impact is limited since the authors already shows how forces lead to channel opening, and this is further teasing apart the forces and motions that are actually the ones that cause the opening.

---

## [Referee Report · Reviewer #3 (Public review)]

Summary:

This manuscript by Duan and Song interrogates the gating mechanisms and specifically force transmission in mechanosensitive NOMPC channels using steered molecular dynamics simulations. They propose that the ankyrin spring can transmit force to the gate through torsional forces adding molecular detail to the force transduction pathways in this channel.

Strengths:

Detailed, rigorous simulations coupled with a novel model for force transduction.

Weaknesses:

Experimental validation of reduced mechanosensitivity through mutagenesis of proposed ankyrin/TRP domain coupling interactions would greatly enhance the manuscript.

---

## [Author Response]

**Reviewer #1 (Public review):**
Summary:This manuscript uses molecular dynamics simulations to understand how forces felt by the intracellular domain are coupled to the opening of the mechanosensitive ion channel NOMPC. The concept is interesting - as the only clearly defined example of an ion channel that opens due to forces on a tethered domain, the mechanism by which this occurs is yet to be fully elucidated. The main finding is that twisting of the transmembrane portion of the protein - specifically via the TRP domain that is conserved within the broad family of channels- is required to open the pore. That this could be a common mechanism utilised by a wide range of channels in the family, not just mechanically gated ones, makes the result significant. It is intriguing to consider how different activating stimuli can produce a similar activating motion within this family. However, the support for the finding can be strengthened as the authors cannot yet exclude that other forces could open the channel if given longer or at different magnitudes. In addition, they do not see the full opening of the channel, only an initial dilation. Even if we accept that twist is essential for this, it may be that it is not sufficient for full opening, and other stimuli are required.Strengths:Demonstrating that rotation of the TRP domain is the essential requirement for channel opening would have significant implications for other members of this channel family.

Thank you for your positive summary and comments.

Weaknesses:The manuscript centres around 3 main computational experiments. In the first, a compression force is applied on a truncated intracellular domain and it is shown that this creates both a membrane normal (compression) and membrane parallel (twisting) force on the TRP domain. This is a point that was demonstrated in the authors’ prior eLife paper - so the point here is to quantify these forces for the second experiment.The second experiment is the most important in the manuscript. In this, forces are applied directly to two residues on the TRP domain with either a membrane normal (compression) or membrane parallel (twisting) direction, with the magnitude and directions chosen to match that found in the first experiment. Only the twisting force is seen to widen the pore in the triplicate simulations, suggesting that twisting, but not compression can open the pore. This result is intriguing and there appears to be a significant difference between the dilation of pore with the two force directions.However, there are two caveats to this conclusion. Firstly, is the magnitude of the forces - the twist force is larger than the applied normal force to match the result of experiment 1. However, it is possible that compression could also open the pore at the same magnitude or if given longer. It may be that twist acts faster or more easily, but I feel it is not yet possible to say it is the key and exclude the possibility that compression could do something similar.

Thank you for your insightful comment. As you pointed out, the membranenormal pushing forces exerted at residues E1571 and R1581 are approximately onethird and two-thirds, respectively, of the membrane-parallel twisting forces. These magnitudes were derived from a previous simulation (Wang et al., 2021), in which we decomposed the resultant force into its membrane-parallel and membrane-normal components upon applying a compressive force to the intracellular AR end. Our results indicated that, upon reaching the TRP helix, the induced twisting force is indeed greater, which partially reflects actual physiological conditions. Therefore, considering the magnitudes of the resultant forces alone, the twisting force is predominantly greater than the pushing force when the AR domain is subjected to compression.

Then the question became, if forces of the same magnitude are applied in either the membrane-normal or membrane-parallel directions, what would the outcome be? To address this, we conducted additional simulations. Considering the situations discussed above, we applied a smaller membrane-parallel force instead of a larger membranenormal force that may disrupt the integrity of protein and membrane structure. As shown in the Figure 2-figure supplement 2, we adjusted the applied membrane-parallel force to either half or one-third of the original value. When we applied half of the force used in the original setup, the channel opened in two out of three trajectories. When applying onethird of the force, the channel opened in one out of three trajectories. Together with our previous results, these findings suggest that if forces of equal magnitude are applied in the membrane-normal and membrane-parallel directions, the membrane-parallel force has a higher probability of inducing channel opening.

Still, one cannot completely exclude the possibility that the pushing force on the TRP helix can open the channel if given a very long time. This becomes unfeasible to examine with MD simulations, so we investigated the likely conformational changes of multiple TRP family proteins upon opening, and found that the TRP rotation is a universal conformational change, while the TRP tilt is much less consistent (Figure 6). These findings gives us more confidence that the twist force plays a more crucial role in channel gating than the pushing force. We have added a new table (Table 1) and a new figure (Figure 6) to present this analysis.

In addition, we did not intend to imply that compression is incapable of contributing to channel opening. In fact, our aim was to highlight that compression can generate both a twisting force and a pushing force, with the twisting force appearing to be the more critical component for facilitating channel opening. We concur that we cannot completely dismiss the possibility that the pushing component may also assist in channel opening. Consequently, we have revised our discussion to enhance clarity.

I also note that when force was applied to the AR domain in experiment 1, the pore widened more quickly than with the twisting force alone, suggesting that compression is doing something to assist with opening.

You are correct that the trajectory corresponding to Experiment 1 (Figure 1-figure supplement 1(b)) indicates pore opening around 300-400 ns, while the trajectory for Experiment 2 (800 ns) shows pore opening around 600 ns. This observation may suggest that the pore opens more rapidly in Experiment 1, assuming that the simulation conditions were identical for both experiments. However, it is important to note that in Experiment 1, an external force was applied to AR29. In contrast, in Experiment 2, the force was applied exclusively to two selected residues on the TRP domain, while other TRP residues also experienced mechanical forces, albeit to a lesser extent. The differing methods of force application in the two experiments complicate the comparison of pore opening speeds under these conditions.

We acknowledge that the compression of the AR spring can facilitate pore opening. This compression generates both a twisting component and a pushing component on the TRP domain. Our simulations and structural analyses of multiple TRP channels suggest that the twisting component plays a predominant role in gating. However, we cannot entirely rule out the possibility that the pushing component may also contribute to this process. We have carefully revised our Result, Discussion and Methods sections to enhance clarity.

Given that the forces are likely to be smaller in physiological conditions it could still be critical to have both twist and compression present. As this is the central aspect of the study, I believe that examining how the channel responds to different force magnitudes could strengthen the conclusions and recommend additional simulations be done to examine this.

Thank you for your valuable comments. We agree that the force applied in Experiment 2 is possible to be larger than the physiological conditions. Therefore, we performed additional simulations to investigate the possibility of opening the pore using smaller torsional forces.

As shown in the new Figure 2-figure supplement 1, we applied half and one-third of the original force and performed three replicate simulations for each condition. With half the force, the pore opened in two out of the three simulations. And with one-third of the applied force, the pore opened in one out of the three replicate simulations. The probability of pore opening within the same simulation time decreased as the applied force was reduced, consistent with our expectations. These new results are provided as supplementary figures in the revised manuscript.

We anticipate that further reductions in the forces will result in additional delays in the opening process; however, this would lead to prohibitive computational costs. Consequently, we have decided to conclude our analysis at this stage and have discussed this matter of the revised manuscript.

The second important consideration is that the study never sees a full pore opening, but rather a widening that is less than that seen in open state structures of other TRP channels and insufficient for rapid ion currents. This is something the authors acknowledge in their prior manuscript in eLife 2021. Although this may simply be due to the limited timescale of the simulations, it needs to be clearly stated as a caveat to the conclusions. Twist may be the key to getting this dilation, but we do not know if it is the key to full pore opening. To demonstrate that the observed dilation is a first step in the opening of pores, a structural comparison to open-state TRP channels would be beneficial in providing evidence that this motion is along the expected pathway of channel gating.

We are grateful for this insightful comment. We acknowledge that our simulations do not capture a fully open state, but rather a dilation that is smaller than the open-state structures of other TRP channels. In our simulations, a pore radius exceeding 2 Å is considered as a partially open state, as this is generally sufficient for the permeation of water molecules or even small cations such as K^+^ and Na^+^ However, the passage of larger molecules and ions, such as Ca^2+^ and clusters of hydrated ions, remains challenging. As you noted, this partial opening may be attributed to the limited timescale of the simulations.

Furthermore, in accordance with your suggestion, we analyzed numerous TRP proteins for which multiple open or intermediate states have been resolved, and we have included a new figure (Figure 6). A clockwise rotation of the TRP domain is observed in the majority of these proteins upon gating. For instance, in the case of RnTRPV1, our analysis revealed that during TRPV1 activation, when different ligands are bound (RTX, DkTX), the pore undergoes gradual dilation, which involves a progressive clockwise rotation of the TRP domain. This analysis provides evidence that the observed motion aligns with expected gating transitions, supporting the notion that twist-induced TRP rotation and pore dilation may represent an initial step in the pore opening process.

Nonetheless, we concur that further studies, including extended simulations, which are currently unfeasible, or experimental validation, will be necessary to ascertain whether our proposed mechanism is adequate for the complete opening of the pore. We have carefully discussed this.

Experiment three considers the intracellular domain and determines the link between compression and twisting of the intracellular AR domain. In this case, the end of the domain is twisted and it is shown that the domain compresses, the converse to the similar study previously done by the authors in which compression of the domain was shown to generate torque. While some additional analysis is provided on the inter-residue links that help generate this, this is less significant than the critical second experiment.

Although experiment three is less significant in revealing the underlying gating mechanism, it provides quantitative measurements of the mechanical properties of the intriguing AR spring structure, which are currently challenging to obtain experimentally. These provide computational predictions for future experiments to validate.

**Reviewer #2 (Public review):**
This study uses all-atom MD simulation to explore the mechanics of channel opening for the NOMPC mechanosensitive channel. Previously the authors used MD to show that external forces directed along the long axis of the protein (normal to the membrane) result in AR domain compression and channel opening. This force causes two changes to the key TRP domains adjacent to the channel gate: (1) a compressive force pushes the TRP domain along the membrane normal, while (2) a twisting torque induces a clock-wise rotation on the TRP domain helix when viewing the bottom of the channel from the cytoplasm. Here, the authors wanted to understand which of those two changes is responsible for increasing the inner pore radius, and they show that it is the torque. The simulations in Figure 2 probe this question with different forces, and we can see the pore open with parallel forces in the membrane, but not with the membrane-normal forces. I believe this result as it is reproducible, the timescales are reaching 1 microsecond, and the gate is clearly increasing diameter to about 4 Å. This seems to be the most important finding in the paper, but the impact is limited since the authors already show how forces lead to channel opening, and this is further teasing apart the forces and motions that are actually the ones that cause the opening.

Thank you for your insightful comments. We appreciate your recognition of our key finding that torque is responsible for increasing the inner pore radius. Indeed, our simulations illustrated in Figure 2 systematically explore the effects of different forces on pore opening. These results demonstrate that membrane-parallel forces are effective, while membrane-normal forces are not within the simulation time. We acknowledge that this study builds upon previous findings regarding force-induced channel opening. However, we believe that further decomposition of the specific forces and motions responsible for this process provides valuable mechanistic insights. By distinguishing the role of torque from the membrane-normal forces of the TRP helix, which is highly conserved across the TRP channel family, our work contributes to a more precise understanding of TRP channel gating. Moreover, in the revised manuscript, we conducted a systematic analysis of the structures of TRP family proteins and discovered that the clockwise rotation of the TRP domain is likely a universal gating mechanism among the TRP family, which significantly enhances and strengthens our original findings (Figure 6).

**Reviewer #3 (Public review):**
Summary:This manuscript by Duan and Song interrogates the gating mechanisms and specifically force transmission in mechanosensitive NOMPC channels using steered molecular dynamics simulations. They propose that the ankyrin spring can transmit force to the gate through torsional forces adding molecular detail to the force transduction pathways in this channel.Strengths:Detailed, rigorous simulations coupled with a novel model for force transduction.

Thank you for your positive comments.

Weaknesses:Experimental validation of reduced mechanosensitivity through mutagenesis of proposed ankyrin/TRP domain coupling interactions would greatly enhance the manuscript. I have some additional questions documented below:

We attempted to measure the mechanical properties of the AR domain and conduct mutagenesis experiments in collaboration with Prof. Jie Yan’s laboratory at the Mechanobiology Institute, National University of Singapore; however, this proved to be a significant challenge at this time. Given the urgency of the publication, we have decided to first publish the computational results and reserve further experimental studies for future investigations.

(1) The membrane-parallel torsion force can open NOMPCHow does the TRP domain interact with the S4-S5 linker? In the original structural studies, the coordination of lipids in this region seems important for gating. In this manner does the TRP domain and S4-S5 linker combined act like an amphipathic helix as suggested first for MscL (Bavi et al., 2016 Nature Communications) and later identified in many MS channels (Kefauver et al., 2020 Nature).

In our analysis of the compression trajectories (trajectory: CI-1, Figure 1-figure supplement 4), we identified stable interactions between the TRP domain and the S4-S5 linker. These interactions primarily involve the residues S1421 and F1422 of the S4-S5 linker, as indicated by the large pink data points in Figure 1-figure supplement 4. Therefore, we agree that the TRP helix and the S4–S5 linker can be considered an amphipathic helical unit, analogous to the amphipathic helix observed in MscL and other mechanosensitive channels. Moreover, the pocket adjacent to the S4-S5 linker has been recognized as a binding site for small molecules in other ligand-activated TRP channels, such as the vanilloid-binding TRPV1. We hypothesize that this unit is likely to play a critical role in the polymodal gating of the TRP channel family, including ligand-induced activation. In the revised manuscript, we have included an analysis of the interaction between the TRP domain and the transmembrane (TM) domain, and we have briefly discussed its implications.

(2) Torsional forces on shorter ankyrin repeats of mammalian TRP channelsIs it possible torsional forces applied to the shorter ankyrin repeats of mammalian TRPs may also convey force in a similar manner?

This is an intriguing question.

To answer your question, we studied the full-length squirrel TRPV1 (PDB: 7LQY, Nadezhdin et al. (2021)) using all-atom steered MD simulations. We applied pushing or torsional forces to the intracellular AR1-2 region of TRPV1, separately (Figure 6-figure supplement 1(a)). Similar to NOMPC, rotation of the TRP domain was observed under both types of mechanical stimulation (Figure 6-figure supplement 1(b-e)). The conformational change induced by the torsional force on the TRP domain resembles the change observed in NOMPC. This suggests that a torsional force applied to the shorter ankyrin repeats of mammalian TRPs may yield similar effects on channel gating. However, given that these ankyrin repeats do not act like tether elements, the implications of these results in the context of biological functions remain unclear. Additionally, in NOMPC, the AR domain is connected to the TRP domain through a linker helix (LH) domain, composed of multiple stacked helices that form a relatively compact structure (Figure 1(a)). In contrast, TRPV1 does not possess a similarly compact LH domain connecting the AR domain to the TRP domain (Figure 6-figure supplement 1(a)). These structural differences render our conclusions regarding NOMPC not directly applicable to TRPV1. We have included an additional discussion about this (Figure 6-figure supplement 1).

(3) Constant velocity or constant forceFor the SMD the authors write "and a constant velocity or constant force". It’s unclear from this reviewer’s perspective which is used to generate the simulation data.

Thank you for pointing out this ambiguity. In our simulations, we first applied constant-velocity pulling to achieve specific force magnitudes, followed by constantforce pulling. This protocol allowed us to initiate the motion of the protein in a controlled manner and observe the response of the system under sustained forces. We have now clarified this in the revised Methods section.

**Reviewer #1 (Recommendations for the authors):**
The language in the paper requires some editing - particularly in the introduction. For example, what is meant by ion channels ’coalescing to form mechanical receptors’? Are the authors implying it requires multiple channels to form a receptor? It is stated that mechanically gated ion channels are only found in nerve endings when in fact they are found in almost every cell type. Another example is the statement ’In the meantime’ the TRP domain was observed to rotate when this observation came prior to the others mentioned before. While these sound like minor edits, they significantly change the meaning of the introduction. I recommend careful editing of the manuscript to avoid accidental inaccuracies like this.

Thank you for your feedback on the clarity and accuracy of the introduction. We have carefully revised the manuscript, particularly the abstract and instroduction sections, to address these concerns:

(1) We have reworded the original sentence ’These mechanosensitive ion channels, coalescing to form mechanical receptors, are strategically positioned within the sensory neuron terminals intricately nestled within the epidermal layer.’ into ’In both vertebrates and invertebrates, mechanosensitive ion channels are widely expressed in peripheral sensory neurons located near or within the surface tissues responsible for detecting mechanical stimuli.’

(2) We have replaced the phrase "In the meantime" with "Interestingly" to introduce the conformational change of the TRP domain that we believe is crucial.

(3) We have carefully reviewed the entire manuscript and used a language editing tool, Writefull integrated within Overleaf, to proof-check the language problems.

**Reviewer #2 (Recommendations for the authors):**
How do the energy values in Figure 3b, compare with the continuum energy values reported by Argudo et al. JGP (2019)? I wonder what value the authors would get with a new replicate run slower - say 200 ns total aggregate simulation? This would probe the convergence of this energy value. It seems important to determine whether the loading velocity of the experiments performed here with the steered MD is slow enough to allow the protein to relax and adopt lower energy configurations during the transition. The true loading is likely to occur on the millisecond timescale, not the nanosecond to low microsecond timescale. That said, I don’t mean to detract from the result in Figure 2, as this is likely quite solid in my opinion given the nearly 1 microsecond simulations and the replicates showing the same results.

Thank you for your valuable suggestions. It is important to note that we calculated different physical quantities compared to those reported in Argudo’s study. In Figure 3b, we calculated the torque (instead of the energy, although they share the same dimensional units) of the long AR bundle (AR9-29 of the four filaments combined) and subsequently determined its torsion coefficient. Argudo’s study calculated the torsional spring constant (𝑘_ɵ_) of three 6-AR-unit stretches of one filament, which were designated as ANK1 (AR 12-17), ANK2 (AR 17-22) and ANK3 (AR 22–27). As the four filaments are coupled within the bundled structure and the torsional axes differ between an individual filament and the four-filament bundle, a direct comparison of the torsional spring constants reported in the two studies is not meaningful.

We agree that extending the simulation time may provide deeper insights into the convergence of energy values. In accordance with your suggestion, we conducted additional simulations to further investigate convergence and compare the results with our existing data, thereby ensuring robustness and consistency. Specifically, we slowed down the original operation of twisting from 10 degrees over 100 ns to 10 degrees over 200 ns, and extended the holding time for selected frames (sampled every 2.5 degrees) from 100 ns to 200 ns. We have updated Figure 3 and relevant main text accordingly. The results of the new simulations are similar to those of the previous ones, with the fitted torsion coefficient revised from (2.31 ± 0.44) × 10^3^kJ mol^−1^ ra^−1^ 1 to (2.30 ± 0.31) × 10^3^ kJmol^−1^ rad^−1^ This close agreement indicates that our simulations are well-converged. Additionally, we updated the compression–twist coupling coefficient, \begin{document}$k_{\mathrm{ct}}=\frac{\Delta L}{\Delta \phi}$\end{document}, from (1.67 ± 0.14) nmrad^−1^ to (1.32 ± 0.11) nmrad^−1^

As you suggested, we conducted an additioanl analysis to determine whether the loading velocity/force with the steered MD is sufficiently slow to facilitate the relaxation of the protein and its adoption of lower-energy configurations during the transition. For simulations involving the application of membrane-normal or membrane-parallel force on the TRP domain, we utilized DSSP (Define Secondary Structure of Proteins) analysis to assess the stability of the secondary structure of the TRP domain. The results indicated that, during the application of external forces, the secondary structure of the TRP domain maintained good stability, as illustrated in Figure 2-figure supplement 3. For simulations involving the rotation of the AR domain, we also analyzed the DSSP of the AR9 to AR11 units, which are positioned directly above the AR8 domain where the twisting force is applied. The secondary structure of the AR domain also exhibited good stability (Figure 4-figure supplement 2). These are briefly discussed in the Methods section of the revised manuscript.

It is unclear to me that the force transmission analysis in Figure 4 provides much insight into the mechanics of opening. Perhaps the argument was made, but I did not appreciate it. Related to this the authors state that the transfer velocity is 1.8 nm/ps based on their previous study. Is this value profound or is it simply the velocity of sound in the protein?

The analysis of force transmission presented in Figure 4 offers detailed insights into the transfer of force along the AR domain. While this may appear straightforward, the information elucidates how a pushing force can induce a twisting force during its transmission through the AR spring structure, as well as the primary contributions that stabilize this transmission pathway. To enhance clarity, we have included an additional discussion.

The force transfer velocity is expected to align with the velocity of sound within the protein. The value of 1.8 nm/ps, however, is specific to the unique structure of the AR spring, which is quite interesting to report in our opinion. Additionally, this rapid transfer speed suggests that the simulation timescale is sufficient for enabling the transfer of compression force from the bottom of the AR domain to the TRP domain in our simulations, given that the simulation timescale is considerably longer than the force propagation timescale within the protein.

The methods description is largely complete, but is missing some details on the MD simulations (barostat, thermostat, piston constants, etc.).

Thank you for pointing out the missing details; we have added the additional information in the revised Methods section.

References

Nadezhdin, K. D., A. Neuberger, Y. A. Nikolaev, L. A. Murphy, E. O. Gracheva, S. N. Bagriantsev, and A. I. Sobolevsky (2021). Extracellular cap domain is an essential component of the trpv1 gating mechanism. Nature communications 12(1), 2154.

Wang, Y., Y. Guo, G. Li, C. Liu, L. Wang, A. Zhang, Z. Yan, and C. Song (2021). The pushto-open mechanism of the tethered mechanosensitive ion channel nompc. Elife 10, e58388.